# microRNAs Are Abundant and Stable in Platelet-Rich Fibrin and Other Autologous Blood Products of Canines

**DOI:** 10.3390/ijms24010770

**Published:** 2023-01-01

**Authors:** Indre Jasineviciute, Md Nazmul Hasan, Juozas Grigas, Arnoldas Pautienius, Arunas Stankevicius, Judita Zymantiene, Naoki Miura

**Affiliations:** 1Department of Anatomy and Physiology, Veterinary Faculty, Lithuanian University of Health Sciences, LT-47181 Kaunas, Lithuania; 2Joint Graduate School of Veterinary Medicine, Kagoshima University, Kagoshima 890-8580, Japan; 3Institute of Microbiology and Virology, Lithuanian University of Health Sciences, LT-47181 Kaunas, Lithuania; 4Joint Faculty of Veterinary Medicine, Veterinary Teaching Hospital, Kagoshima University, Kagoshima 890-8580, Japan

**Keywords:** platelet-rich fibrin, microRNAs, canine

## Abstract

Various microRNAs (miRNAs) present in autologous blood products of canines have not been studied recently. We aimed to elucidate the existence of miRNAs in platelet-rich fibrin (PRF) and the stability of canine autologous blood products under various storage conditions. Total RNAs were isolated from PRF and other autologous blood products following newly adapted protocols used in commercial kits for plasma and tissue samples. Quantitative real-time polymerase chain reaction analysis (qPCR) was used to detect miRNAs in autologous blood products. The miR-16, miR-21, miR-155, and miR-146a were abundant in PRF and other autologous blood products of canines. Furthermore, we found they could maintain stability under protracted freezing temperatures of −30 °C for at least one month. Our findings revealed that PRF might be a stable resource for various canine miRNAs.

## 1. Introduction

Autologous blood products have been widely used in various medical fields [1,2,3]. At least several studies have demonstrated that PRF has the potential to accelerate the wound-healing process [4,5]. A three-dimensional fibrin matrix in PRF attracts and traps significant quantities of T cells, B cells, macrophages, monocytes, dendritic cells, and growth factors [4]. However, the usefulness of PRF for canine patients has not been sufficiently investigated. Moreover, other autologous blood plasma products have also been previously explored as potential tools for accelerating wound healing [6,7].

Short and compact non-coding RNA molecules called microRNAs (miRNAs) are widely researched in many immunology and oncology studies [8,9,10]. It is known that miRNAs expression patterns and regulation play a significant role in wound healing and inflammation stages. Apoptosis, cell division, and migration are the main functions of the miRNAs [11]. Chronic wounds with delayed healing, particularly in diabetic patients, are attributed mainly to miRNA dysregulation [12]. In particular, miR-16, miR-21, miR-155, and miR-146a are crucial for innate immune response and the first stages of the wound healing process [12,13,14]. Moreover, miR-146a responds strongly to microbial infection, while miR-16 is a crucial factor in hemostasis and is involved in the mechanisms of tumor development [12]. The effect of miR-146a is based on the TNF signaling pathway, which targets IRAK1 and TRAF6 and inhibits inflammation [15]. Its ability to reduce oxidative stress and inflammation is the crucial factor that leads to effective wound healing [12].

Human research has previously demonstrated total RNA isolation from PRP and other blood plasma products [16]. On the other hand, a detailed protocol of the whole RNA isolation from PRF of canines has not been described yet. To our knowledge, only one animal study analyzed miRNA in the PRF of rabbits [17]. Moreover, the stability of miRNAs has been previously described, but this aspect has not been examined in canine blood products [16].

The authors hypothesized that miRNA might be one of the components in canines’ autologous blood products that have not been detected yet. On the other hand, their successful isolation is a challenging part of the study. Comparing the presence of miRNAs of PRF to platelet-rich plasma (PRP) and platelet-poor plasma (PPP), which were also explored as potential tools for accelerating wound healing [6,18], was the main goal of the present study (Figure 1). The additional aim of our research involved the analysis of PRF sample preparation for total RNA isolation, which is a mandatory step to reach the primary goal. In this investigation, miRNA abundance measurements from different canine autologous blood products were analysed over one week and one month. It’s important to mention, that the stability of miRNAs after prolonged freezing conditions is a crucial feature that can be beneficially used during PRF adaptation for clinical use as a miRNA carrier to the wound. Although miRNA detection and comparison in PRF and other autologous blood products was the main focus of the study, additional analysis of the sample preparation technique and miRNA stability check helped us to gain more practically adaptable results.

## 2. Results

### 2.1. miRNAs Are More Abundant in PRF than Other Autologous Blood Products of Canines

Synthetically spiked-in cel-miR-39 group was used for miRNA comparison (Figure 2A). Results showed that miR-16 is more abundant in PRF clots compared to PRP with a mean rank difference of Ct values (MRD) = −24.4, *p* < 0.001, as well as PPP-PRF (MRD = −38.8, *p* < 0.001) and PPP-PRP (MRD = −23, *p* < 0.01) (Figure 2B). Both components of PRF clot, including fibrin and secretome, had similar Ct values (MRD = −11.8) in qPCR analysis. The difference in miRNAs between PRF liquid and fibrin parts was insignificant. The difference between PRF clot and PRP samples has shown a strong significant difference with regard to the Ct value of miR-21 (MRD = −27.3, *p* < 0.0001) (Figure 2C). Significant miR-21 difference between PRF and PPP samples made of both PRP (MRD = −19.9, *p* < 0.01) and PRF (MRD = −25.5, *p* < 0.0001) has also been observed.

Similar levels of significance of miR-146a were found in the analysis between PRF and PRP (MRD = −28.5, *p* < 0.001), PPP-PRF (MRD = −39.08, *p* < 0.0001), and PPP-PRP (MRD = −24.87, *p* < 0.01) (Figure 3A). Meanwhile, the result of miR-155 showed the same trend where the Ct value between PRF and other autologous blood products was significant: PRF and PRP (MRD = −21.2, *p* < 0.01), PRF and PPP-PRF (MRD = −35.2, *p* < 0.001), and PRF and PPP-PRP (MRD = −22.8, *p* < 0.01) (Figure 3B). In contrast, they were less abundant than miR-16 and miR-21 (Figure 2). The results show a higher Ct value in platelet-poor products when comparing platelet-rich and platelet-poor blood products.

### 2.2. miRNAs Are Stable in Different Autologous Blood Products of Canines after Frozen at −30 °C

The total RNA isolation technique was adapted to evaluate the qPCR results of miR-16 and miR-21 after keeping samples in frozen conditions (−30 °C) for one week and one month (Figure 1). The results revealed no statistically significant difference between all samples of PRF, PRP, and PPP following an extended time of cold conditions before collecting the total RNA. Ct value is stable after measuring miR-16 (Figure 4A) and miR-21 (Figure 4B) Ct value after one week and one month freeze of samples in −30 °C temperature (Figure 4).

### 2.3. miRNAs Expression Status of Different Autologous Canine Blood Products

Our study revealed that miR-16, miR-21, miR-146a, and miR-155 could be successfully isolated from PRF, PRP, and PPP products made of canine blood using standard whole RNA isolation kits.

The results of the qPCR analysis showed that a high number of miRNAs are detected in every sample of healthy canines, which is reflected by the cycle threshold value (Ct). It was observed that there is no significant difference in isolating RNA from the fibrin or secretome parts of the PRF clot using different RNA isolation kits (Figure 2 and Figure 3).

Correlation analysis of miRNAs showed that a strong positive correlation exists between miR-155 and miR-16 in PPP-PRF (*p* < 0.01) and PPP-PRP samples (*p* < 0.05) (Figure 5C,D). The co-existing tendency of both miRNAs that regulate B cell proliferation confirms their existence in PPP-PRP and PPP-PRF products. Moreover, miR-21 had a strong positive correlation in PPP-PRP samples with miR-16 and miR-155 (*p* < 0.01; *p* < 0.05). A robust positive correlation detected in PPP-PRF samples was between miR-16 and miR146a (*p* < 0.001) as well as miR-155 and miR-146a (*p* < 0.001). Furthermore, strong positive correlations were present in the PPP-PRP sample between miR-146a and miR-21 (*p* < 0.0001), as well as miR-146a and miR-16 (*p* < 0.01) and miR-155 (*p* < 0.05). Differences are based on various patients’ characteristics. If not otherwise stated, no significant correlation result was found in platelet-rich products; the detailed correlation results are represented in Figure 5.

## 3. Discussion

Different autologous blood products were analysed recently to evaluate extracellular miRNA presence and its impact on the immunity processes in humans and laboratory animals [16,17,19]. This study aimed to adapt current methods of miRNAs analysis to a new study object: PRF of dogs. Furthermore, we attempted to examine the impact of the extended freezing time of PRF and its effect on the abundance of different miRNAs. Commercial kits have been widely used for RNA isolation from different substances of samples for a long time. In the present study, we successfully applied the standard protocol for RNA isolation from the PRF fibrin and secretome parts. Separate kits for complete RNA isolation were used to prepare PRF samples of two other substances: liquid and fibrin. Additional steps, including PRF separation into two substances and fibrin part preparation, helped us isolate RNA for the first time. Separating fibrin from the liquid part and cutting it into smaller parts before adding homogenate additive and mixing multiple times were key steps for RNA isolation from the PRF fibrin part. Adapting the standard protocol might be a valuable tool for future investigation of different autologous blood products and their effectiveness.

Recent findings revealed that PRP has a higher level of different miRNAs than PPP in humans [20]. In contrast, we could not find statistically significant differences in analyzing various miRNAs in PRP and PPP in dogs. The result might be impacted by specific kinds of miRNAs chosen for the study, while it is well known that miRNA levels are closely related to platelet expression [21]. On the other hand, to our knowledge, miRNA levels in dogs have not been investigated yet. Higher levels of miRNA-16, miRNA-21, miRNA-146a, and miRNA-155 were detected in PRF samples compared to PRP and PPP in the present study. Our data is consistent with previous findings of higher mRNA levels of injectable PRF compared to human PRP samples [18].

miR-16 has the highest level of abundance among all other miRNAs, especially in PRF. This tendency might be because miR-16 is highly expressed in the plasma erythrocytes [8]. Its biological function includes angiogenesis by targeting VEGFR2 and FGFR1 in endothelial cells and regulating B lymphocytes [22,23,24]. miR-16 presence in PRF and other autologous blood products in this study supports the beneficial effect for angiogenesis activation that was detected in previous research as one of the main functions of regenerative materials [25].

The primary regulators of the inflammatory response: miR-146a, miR-155, and miR-21, were detected in PRF and other autologous blood products. The high abundance of miR-146a in PRF supports its beneficial use as a material that can deliver miR-146a to a chronic wound. A previous study revealed that the downregulation of miR-146a is related to the development of diabetic wounds, and its delivery leads to better chronic wound healing [12,26]. Moreover, the beneficial use of PRF in diabetic wound treatment was already noticed [27]. Furthermore, miR-155 is expressed in activated macrophages and maturing B cells and creates the balance of immunity response with miR-146a [28]. This function might be related to PRF’s inflammatory macrophage activation and is already supported by a previous study [29]. Even though the miRNAs abundance level was the highest in PRF, we cannot conclude that PRF has the most significant number of miRNAs in its composition due to the lack of expression-value analysis, which could not be completed without a comparison group of diseased canines.

Despite that, we can assume that PRF has less interference in isolating miR-16, miR-146a, miR-155, and miR-21. A large number of WBC leads to the idea of a higher number of miRNAs in PRF, but they might be left undetected in the extracellular vesicles [30].

It is known that synergetic features of miRNAs support different immunity functions, like regulating genes in metabolic processes [31]. Significant correlation of various miRNAs in our research might suggest the inter-dependency of miRNAs. Every evaluated miRNA plays a role in the mechanism of inflammation. It implies the idea of evaluating miRNAs not independently but rather as a whole background of their cooperation.

The stability of miRNA in different autologous blood products was discussed in human research. Authors demonstrated that miRNAs could be extracted even after being subjected to various freezing conditions [16]. Our study supports miRNA stability in different blood products of canines. Moreover, the same result could be confirmed in PRF, which has not been previously analyzed. Extended freezing conditions for one week or one month have not consistently affected the count of miRNAs in all canine blood products. miRNAs can stay stable after platelet removal and multiple freezing times. This result shows a high potential for miRNA adaptability to be used as a biomarker in dogs.

Furthermore, the analysis has the potential to be expanded because of current research limitations that cannot confirm the count of miRNAs completely. Another drawback of the study is a lack of comparison between the control group and patients with persistent inflammation or neoplasia that could be used for expression-value analysis.

## 4. Materials and Methods

All tests of canines met ARRIVE welfare and ethical guidelines for animal research. Blood samples were collected from dogs receiving treatment at the Kagoshima University Veterinary Teaching Hospital with the ethically approved number KVH220001 (Kagoshima, Japan). Twenty samples of various autologous blood products (including PRF, PRP, platelet-poor plasma made from PRF (PPP-PRF), and platelet-poor plasma produced from PRP (PPP-PRP) were used in our investigation, which was obtained from five clinically healthy adult dogs (*n* = 5), which was the total number of animals used in our study (Table 1). The sample number was chosen by detecting the minimal number of samples needed for hypothesis support.

### 4.1. The Whole Blood Sample Collection and Preparation of PRF, PRP, and PPP

The whole blood sample was collected from the jugular vein into a sterile 4.5 mL glass tube with sodium citrate and stored at 4 °C for further analysis. PRP was obtained by centrifugation 800× *g*-forces for 5 min at room temperature, and 0.6 mL of PRP sample was collected into plain tubes and stored at −30 °C temperature. The remaining part of PRP was centrifuged again at 2300 g-force for 5 min at room temperature, collected into plain tubes, and stored at −30 °C.

For the preparation of PRF, another 8 mL of autologous blood sample was collected from the jugular vein into a 10 mL sterile glass tube without anticoagulant and immediately centrifuged within one minute. PRF was obtained by a fixed-angled centrifuged at using 700× *g*-forces for 12 min at room temperature.

The upper liquid part of PPP from the PRF tube was collected into plain tubes, and the PRF sample was gently separated by a scalpel blade 2 mm below the connection to the bottom layer of red blood cells. PRF sample was gently ground using a dull instrument until the fibrin and liquid parts were entirely separated. PRF and PPP from PRF samples were stored at −30 °C temperature. The fibrin sample from PRF was split into smaller pieces before freezing and storing. All samples were used for RNA extraction 1–7 days after freezing. The preparation of different autologous blood products is represented in Figure 1.

### 4.2. RNA Extraction

A mirVana RNA Isolation Kit (Thermo Fisher Scientific, Waltham, MA, USA) was used to isolate total RNA from fibrin. A mirVana PARIS Kit (Thermo Fisher Scientific, Waltham, MA, USA) was used to isolate RNA from 200 μL of plasma according to the manufacturer’s instructions. Briefly, recommended volume of miRNA homogenate additive (mirVana,) or 2X Denaturing Solution (PARIS) was added to each sample and mixed by vortexing. After adding 2X Denaturing Solution, plasma samples were spiked with synthetic cel-39 and incubated on ice. An equal volume of acid-phenol: and chloroform (Ambion) was added to each aliquot. The upper aqueous phase solutions were separated and mixed with 1.25 volumes of 100% molecular-grade ethanol. The solution was passed through a mirVana or PARIS column according to the manufacturer’s instructions. Finally, 100 µL of preheated 95 °C elution solution was used to elute total RNA. The concentration of total RNA isolated from fibrin was quantified using a NanoDrop 2000c spectrophotometer (Thermo Fisher Scientific, Waltham, MA, USA) [32,33,34].

### 4.3. Quantitative Real-Time PCR

For plasma, an equal volume of 1.25 μL of total RNA and 4.5 μL synthetic cel-39 was spiked into the plasma samples and measured as an internal control to confirm equivalent RNA isolation. Then, it was reverse transcribed into cDNA using the TaqMan MicroRNA Reverse Transcription Kit (Thermo Fisher Scientific, Waltham, MA, USA) according to the manufacturer’s protocol.

TaqMan First Advanced Master Mix Kit and QuantStudioTM 3 real-time PCR System (Thermo Fisher Scientific, Waltham, MA, USA) was used for qRT-PCR. Thermal cycling was carried out following the manufacturer’s instructions. All experiments were carried out in duplicates. The 2^−ΔΔCT^ method was used to measure the miRNA expression [34,35]. To measure relative expression, cel-miR-39 was used as an internal control in plasma samples [36]. The TaqMan miRNA assays used for qPCR in this study were: cel-miR-39 (ID: 478293), miR-16 (ID: 000391), miR-21 (ID: 000397), bta-miR-146a (ID: 005896), and miR-155 (ID: 002287).

### 4.4. Statistical Analysis

GraphPad Prism 9 was used for statistical analysis. The Kruskal–Wallis test was performed to assess differences between different characteristic subgroups. Spearman r correlation test was performed to find relations between different types of miRNAs. The differences were considered significant when the *p*-value was <0.05 (* *p* < 0.05, ** *p* < 0.01, *** *p* < 0.001).

## 5. Conclusions

In this study, we successfully detected four different miRNAs typically associated with roles in inflammatory and oncological processes, namely, miR-16, miR-21, miR-146a, and miR-155, in all tested autologous blood products. The authors concluded that various miRNAs are the most abundant in PRF compared to PRP and PPP, and the result suggests that the miRNA isolation process is more accessible in PRF. Even though blood cells are sensitive to low-temperature conditions, the remarkable stability of miRNAs in frozen autologous blood products confirms their high potential to be used in clinical therapy.

## Figures and Tables

**Figure 1 ijms-24-00770-f001:**
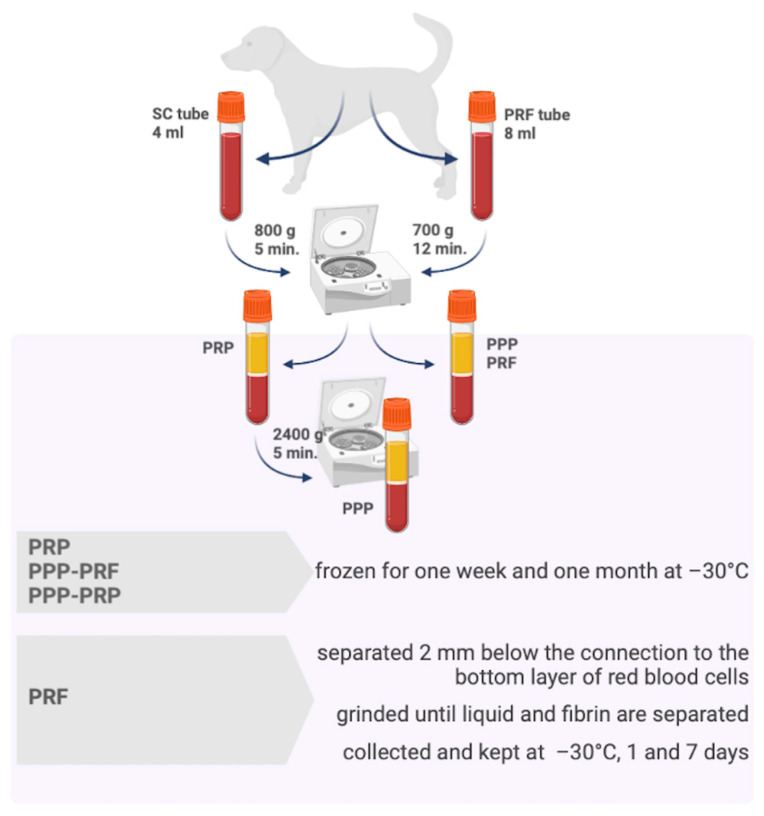
Preparation of platelet-rich fibrin and other autologous blood products of canine. The scheme was created using BioRender.

**Figure 2 ijms-24-00770-f002:**
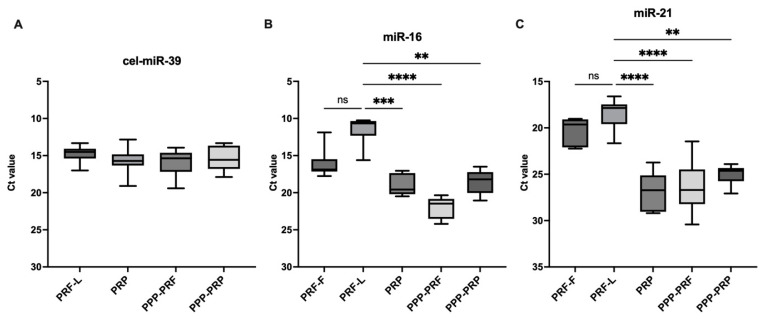
Ct value levels of miR-16 (**B**) and mir-21 (**C**), as well as synthetically spiked-in cel-miR-39 (**A**) in different autologous blood products of canine, qPCR analysis results, are represented as Ct value. After the first collection day, total RNA was isolated from the samples and kept in −30 °C. The PRF clot sample was separated into fibrin (PRF-F) and secretome (PRF-L) substances. The level of the whole RNA in PRF-F was measured with a spectrophotometer. Significance is marked ** *p* < 0.01, *** *p* < 0.001, **** *p* < 0.0001, ns—non significant.

**Figure 3 ijms-24-00770-f003:**
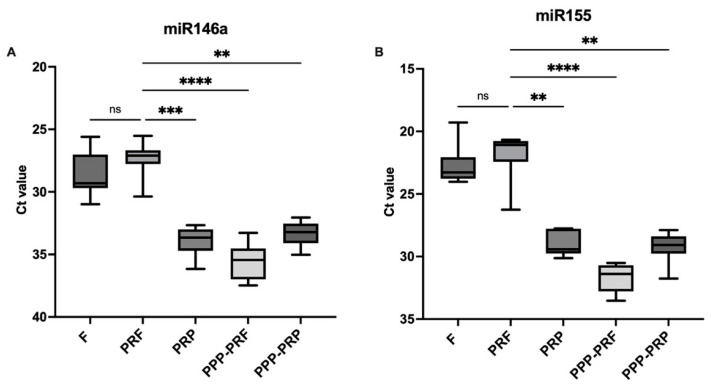
Comparison of Ct value levels of miR-146a (**A**) and mir-155 (**B**) in different autologous blood products of canine. qPCR analysis results are represented as Ct value. After the first collection day, total RNA was isolated from the samples and kept in −30 °C. PRF clot sample was separated into two substances: fibrin (PRF-F) and secretome (PRF-L), level of RNA in PRF-F was measured with a spectrophotometer. Significance is marked ** *p* < 0.01, *** *p* < 0.001, **** *p* < 0.0001, ns—non significant.

**Figure 4 ijms-24-00770-f004:**
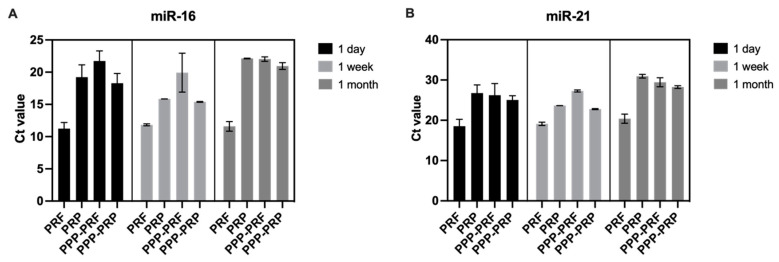
miR-16 (**A**) and miR-21 (**B**) are stable in different autologous blood products of canines. Samples were frozen at −30 °C for one week and one month before whole RNA collection. The graph represents the Ct value of qPCR analysis results with mean and standard deviation.

**Figure 5 ijms-24-00770-f005:**
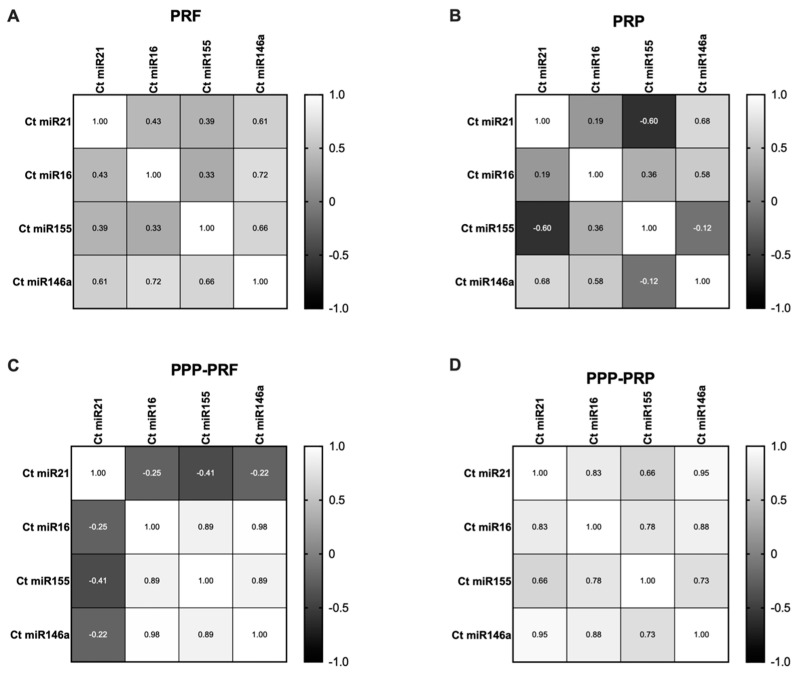
Representation of Spearman r correlation test variety among different autologous blood products. The Ct value of other miRNAs in PRF (**A**), PRP (**B**), PPP-PRF (**C**), and PPP-PRP (**D**) was compared.

**Table 1 ijms-24-00770-t001:** Characteristics of the dogs.

Variables	Dogs (*n* = 5)
Age, y/o	
0–8 (adult)	5
9–17 (senior)	0
Sex	
Female, *n* (%)	3 (60)
Male, *n* (%)	2 (40)
Weight, kg	
0–10, *n* (%)	1 (20)
11–20, *n* (%)	4 (80)
Breed	
Mix	3
Shiba	2

## Data Availability

The data presented in this study are available in this article and can be found at https://doi.org/10.5281/zenodo.7299985? (accessed on 8 November 2022).

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
