# Peer review of "microRNAs Are Abundant and Stable in Platelet-Rich Fibrin and Other Autologous Blood Products of Canines"

_ijms, 2023, doi:10.3390/ijms24010770_

Round 1
Reviewer 1 Report
This manuscript covers an interesting topic and a new study. The authors did a great job, but in several sections / points they need to carefully improve the manuscript.
- In line 19, first mention the full name of the PCR and then use its acronym.
- Also, in line 34 it enhances the sentence by adding before the full name of the miRNAs.
- Perhaps the paragraph in lines 54 to 62, should be revised and reorganized better starting with the main aim and then listing their additional objectives. As expressed, it does not clarify the study.
- At point 2.1 the descriptions of Figure 1 and Figure 2A are missing.
- At the end of the discussion section, authors can mention the limitations of their study.
- The Materials and Methods section should be reorganized into subsections respecting the subsections of the results and ending with the subsection called “Statistical Analysis”, where the description should be separate.
Author Response
We are grateful for the reviewers' comments and opinions and would like to present our concise overview
Reviewer 1
The changes we made include these aspects:
- Correction of acronyms usage, including full names in lines 19 and 34.
- Revision and reorganisation of paragraph lines 54-62 to clarify our primary goal of the study.
- Correction of Figures descriptions as suggested. Figure 1 explanation is included in Introduction and the Materials and Methods part.
- The end of the discussion has included our study limitations already, but we expanded and made it more precise and understandable.
- The materials and Methods part was reorganised and separated into subsections as suggested.
Reviewer 2 Report
In this study, we successfully detected four different miRNAs typically associated with roles in inflammatory and oncological processes: miR-16, miR-21, miR-146a, and miR- 155 The authors concluded that various miRNAs are the most abundant in PRF compared to PRP and PPP. The result suggests that the miRNAs isolation process is easier in PRF. Even though blood cells are sensitive to low temperature conditions, the remarkable stability of miRNAs in frozen autologous blood products confirms their high potential to be used in clinical therapy.
The topic is interesting and well written.
Null and alternative hypotheses should be added
You must follow the guide criteria arrive ple size relates to the number of experimental units in each group at the start of the study, and is usually represented,… porting the characteristics of all animals used …………….
Author Response
We are grateful for the reviewer's comments and opinions and would like to present our concise overview
Reviewer 2
The changes we made include these aspects:
- We added a more detailed hypothesis description to the manuscript.
- Modification of Methods section adding a statement about following ARRIVE guidelines and adding more information about our patients.
- Correction of Table 1, including more clarified information about patients' characteristics.
Round 2
Reviewer 2 Report
Table 1. Characteristics of the patients. replace by characteristics of the animals / dogs / healthy adult dogs
Author Response
Thank you for your additional suggestion.
The title of Table 1 was changed following the reviewers' opinion.